# Genetically Encoded Self-Assembling Protein Nanoparticles for the Targeted Delivery In Vitro and In Vivo

**DOI:** 10.3390/pharmaceutics15010231

**Published:** 2023-01-10

**Authors:** Anastasiia S. Obozina, Elena N. Komedchikova, Olga A. Kolesnikova, Anna M. Iureva, Vera L. Kovalenko, Fedor A. Zavalko, Tatiana V. Rozhnikova, Ekaterina D. Tereshina, Elizaveta N. Mochalova, Victoria O. Shipunova

**Affiliations:** 1Moscow Institute of Physics and Technology, 141701 Dolgoprudny, Russia; 2Nanobiomedicine Division, Sirius University of Science and Technology, 354340 Sochi, Russia

**Keywords:** protein nanoparticles, targeted delivery, self-assembly, transferrin, lactoferrin, casein, encapsulin, albumin, lumazine synthase, magic bullet

## Abstract

Targeted nanoparticles of different origins are considered as new-generation diagnostic and therapeutic tools. However, there are no targeted drug formulations within the composition of nanoparticles approved by the FDA for use in the clinic, which is associated with the insufficient effectiveness of the developed candidates, the difficulties of their biotechnological production, and inadequate batch-to-batch reproducibility. Targeted protein self-assembling nanoparticles circumvent this problem since proteins are encoded in DNA and the final protein product is produced in only one possible way. We believe that the combination of the endless biomedical potential of protein carriers as nanoparticles and the standardized protein purification protocols will make significant progress in “magic bullet” creation possible, bringing modern biomedicine to a new level. In this review, we are focused on the currently existing platforms for targeted self-assembling protein nanoparticles based on transferrin, lactoferrin, casein, lumazine synthase, albumin, ferritin, and encapsulin proteins, as well as on proteins from magnetosomes and virus-like particles. The applications of these self-assembling proteins for targeted delivery in vitro and in vivo are thoroughly discussed, including bioimaging applications and different therapeutic approaches, such as chemotherapy, gene delivery, and photodynamic and photothermal therapy. A critical assessment of these protein platforms’ efficacy in biomedicine is provided and possible problems associated with their further development are described.

## 1. Introduction

Nanoparticles of different origins, possessing unique intrinsic properties, open up endless possibilities in the diagnosis and treatment of socially significant diseases [1,2,3]. A large number of nanoparticles of different compositions, including organic (liposomes, exosomes, polymer particles, dendrimers, carbon nanomaterials, polymeric micelles) and inorganic (iron oxide, gold, silver, TiN, quantum dots), have been developed and thoroughly studied [4,5,6,7,8,9,10]. However, only a limited number of drug formulations within the composition of nanoparticles have reached clinical applications. At the same time, more than three hundred protein-based medications have been approved for clinical use [11] (there are more than one hundred medications of therapeutic antibodies only [12,13,14]; a great variety of diagnostic antibodies, including those for COVID-19 diagnostics [15]; therapeutic proteins such as different enzymes [16]; and therapeutic peptides [11]), which indicates that proteins are easier to translate into clinical practice in comparison with nanoparticles. The issues of entering the biomedical market of protein medications, concerning the standardization techniques and batch-to-batch reproducible synthesis methods, are much less pronounced in comparison with solid nanoparticles. The protein sequence is encoded in DNA, which results in the protein being produced in only one possible way. Along with globular proteins, protein-based self-assembling nanoparticles are reproducibly generated in bacterial producers with high yields [17]. To date, many types of protein self-assembling nanoparticles are known and many of them have confirmed their efficacy as targeted drug carriers in vitro and in vivo (Figure 1).

Moreover, using genetic engineering manipulations, such self-assembling protein nanoparticles can be modified with targeted recognizing molecules such as antibodies and their derivatives, namely, antibody mimetics—DARPins or affibodies, as well as various targeted peptides [18,19,20,21]. Such methods make it possible to obtain self-assembled nanoparticles for targeted delivery to cancer cells in order to minimize side effects, reduce administered doses of chemotherapeutic drugs, and minimize systemic toxicity, thus realizing a “magic bullet” concept. Since the “magic bullet” idea was initially formulated by Paul Ehrlich [22,23], a number of approaches have been designed for targeted drug delivery (TDD), and most of them are based on nanoparticle carriers [24]. However, despite the endless number of studies directed toward the development of TDD systems, only a limited number of targeted particle-based candidates have entered clinical trials, with no FDA-approved **targeted** nanoparticles existing today [25]. Insufficient efficiency and low bath-to-batch biotechnological reproducibility hamper the development of targeted solid nanoparticle-based TDD systems.

We believe that the combination of the huge biomedical potential of protein carriers as nanoparticles and the standardized protocols of protein purification will make significant progress in magic bullet creation possible, bringing modern biomedicine to a new level.

In this review, we are focused on the currently existing platforms for self-assembling protein nanoparticles for creating new-generation nanomedicines based on transferrin, lactoferrin, casein, lumazine synthase, albumin, ferritin, and encapsulin proteins, as well as on proteins from magnetosomes and virus-like particles (Figure 1). The mentioned self-assembling proteins were used for the design of TDD systems for bioimaging purposes and different therapeutic approaches, such as chemotherapy, gene delivery, photodynamic therapy, or photothermal therapy.

## 2. Protein-Based Targeting Self-Assembling Nanoparticles for Biomedical Applications

### 2.1. Targeted Ferritin Nanoparticles

Ferritin is a 450 kDa self-assembled spherical protein that has been frequently used for bioimaging and as a platform for targeted drug delivery [26,27,28,29,30]. It was shown that 24 protein subunits assemble into nanoparticles with an outer diameter of 12 nm [31,32,33]. The inner diameter of the ferritin nanoparticle was found to be 8 nm, which allows a significant number of therapeutic molecules to be loaded inside. The most common chemotherapeutic drugs loaded into ferritin nanoparticles are doxorubicin [34,35], paclitaxel [36,37], cisplatin [38,39,40], and curcumin [41,42,43,44,45]. Mammalian ferritin is composed of an H-chain (21 kDa) and an L-chain (19 kDa), which are self-assembled with each other in various proportions into nanoparticles. Currently, ferritins derived from various species are used for targeted drug delivery, for instance, horse spleen ferritin, human ferritin, and ferritin from *Archaeoglobus fulgidus* [46]. One of the most widely used types of ferritin nanoparticles for targeted drug delivery is H-chain ferritin (HFn) consisting of only the H-chain. Since ferritin is present in almost all living organisms, including humans, for iron storage, and does not contain toxic elements, ferritin nanoparticles do not provoke any immune responses [33,47].

H-chain ferritin has been shown to specifically recognize and bind to the transferrin 1 receptor (TfR1) on the surface of cells [48]. TfR1 is frequently overexpressed on cancer cells, and consequently, to achieve specific binding to this receptor, the surface of ferritin nanoparticles does not require any modifications. Another subunit of ferritin, the L-chain, has been found to specifically interact with scavenger receptor class A member 5 (SCARA5) which is upregulated in breast cancer stem cells [43,49]. This feature makes ferritin a convenient platform for targeted drug delivery to TfR1- or SCARA5-overexpressing cancer cells.

The natural targeting ability of ferritin has been successfully applied in the development of targeted drug delivery systems for cancer treatment. HFn loaded with doxorubicin (HFn-Dox) was used for in vitro delivery to TfR1-overexpressed HT-29 human colon cancer cells. HFn-Dox was labeled with Cy5.5 and incubated with HT-29 cells. At 72 h post-incubation, tumor cell apoptosis was revealed by employing fluorescence microscopy. A single-dose injection of HFn-Dox into HT-29 tumor-bearing mice resulted in tumor growth inhibition: the tumor volume was reduced below 100 mm^3^ within 20 days. Compared to groups treated with Dox, HFn, and PBS, tumor volume exceeded 1000 mm^3^ by day 18 [34]. In a series of works, ferritin nanoparticles with unmodified surfaces were used to treat breast cancer [41,43,50,51,52]. For example, horse spleen apoferritin delivered 97 µg/mL of encapsulated curcumin to the MCF-7 cell line through the interaction of L-chain with SCARA5. Aside from curcumin, nanoparticles were also loaded with the MRI contrast agent (Gd-HPDO3A), allowing the determination of the amount of delivered curcumin [45]. In the succeeding work, the nanoparticles inhibited the growth of breast tumors in mice in 62.5% of cases [43]. Alongside breast cancer treatment, ferritin nanoparticles are used to specifically deliver drugs to gastric cancer cells. For example, doxorubicin-loaded HFn treatment led to a tumor growth inhibition (TGI) of 91.1% in TfR1-positive patient-derived xenograft models of gastric cancer [53].

Nonetheless, the surface of ferritin nanocages could be modified through either genetic engineering or chemical conjugation techniques to enhance their ability to specifically target cells or provide dual-targeting ability. Particularly, using genetic engineering, different proteins such as antibodies or peptides could be introduced on the surface of nanoparticles. For example, the dual-targeting ability was achieved in HFn nanoparticles genetically fused with an integrin α2β1 targeting ligand (2D-HFn). Unlike most drug platforms, HFn can cross the blood–brain barrier (BBB), whereas the integrin α2β1 targeting ligand specifically binds to integrin α2β1 overexpressed in glioma (Figure 2) [54,55,56]. As a result, the flow cytometry test demonstrated that fluorescently labeled with fluorescein isothiocyanate (FITC) 2D-HFn particles traversed the BBB and bound to U-87MG cells. The same effect was achieved in vivo, which was analyzed by MRI and IVIS imaging of IRDye-800-2D-HFn injected into orthotopic tumor-bearing animals. Doxorubicin-loaded 2D-HFn caused the death of 69% of U-87MG cells compared to 29% of cells that died from free doxorubicin. After three weeks of Dox-loaded 2D-HFn treatment on U-87MG tumor-bearing mice, the average tumor weight was less than 0.2 g. In the orthotopic glioma brain tumor model, tumor volume after treatment with Dox-loaded 2D-HFn reached 7.53 ± 3.16 mm^3^ [57].

Glucose-regulated protein (GRP78) is a cell surface receptor on tumor cells, including hepatocellular carcinoma (HCC) cells. GRP78 overexpression is associated with tumor resistance to apoptosis and chemotherapeutical drugs [58,59]. Genetically modified *Pyrococcus furiosus* ferritin Fn (HccFn) exposing the SP94 peptide that specifically binds to GRP78 on HCC cells was found to accumulate up to 400 doxorubicin molecules per HccFn (HccFn-Dox). FITC-labeled HccFn-Dox specifically bound to HCC cells, which was indicated by confocal laser scanning microscopy. The flow cytometry test confirmed the binding efficiency of this protein to cells with an affinity constant of less than 1 nM. In vivo studies showed that tumors were eliminated in four of six HccFn-Dox-treated mice with HCC tumors [35].

Photothermal therapy (PTT) involves using photosensitizer agents that initially absorb irradiated light and then convert it into heat. Generated heat destroys cancer cells by inducing apoptosis [60]. PTT has been successfully combined with ferritin nanoparticles carrying photothermal agents to prevent the damaging of normal cells. HFn carrying ultrasmall CuS (HFn-CuS) under 808 nm laser irradiation caused the death of about 70% of human glioblastoma U87MG cells. HFn-CuS was injected into human glioblastoma U87MG-bearing Nu/Nu mice with subsequent 808 nm laser exposure. It resulted in 100% tumor elimination without obvious systemic toxicity [61]. PTT could also be used with photodynamic therapy (PDT). PDT involves photosensitizing agents activated by light to produce reactive oxygen species, e.g., singlet oxygen that damages tumor cells. For example, sinoporphyrin sodium-loaded HFn was functionalized with RGB peptide (R-HFn-DVDMS) that can specifically bind to integrin αvβ3. In vitro studies showed that the relative cell viability of 4T1 cells was less than 40% at 40 µg/mL of DVDMS from R-Fn-DVDMS induced PTT + PDT. In 4T1 tumor-bearing mice, R-HFn-DVDMS treatment with 630 nm laser exposure fully eliminated the tumor in 2 weeks [62], thus confirming the great biomedical potential of ferritin nanoparticles for cancer treatment.

### 2.2. Targeted Transferrin Nanoparticles

Transferrin (Tf) is a 75.2 kDa glycoprotein. Tf can bind to its receptor TfR, thus triggering cellular endocytosis. TfR is abundantly expressed in several human tumors, therefore, Tf has been widely used as a targeting molecule exposed on the surface of drug carriers over the past 20 years [63,64,65,66]. Transferrin nanoparticles (Tf NPs) were reported to be synthesized through a self-assembling method in the range of 5 to 200 nm [67]. Hence, with the advantage of natural targeting ability, Tf NPs can specifically deliver drugs to TfR-overexpressed tumors. Transferrin is not immunogenic because it is naturally present in the human body for iron delivery [68]. The main disadvantage of this targeted drug delivery system is that glycoprotein production in prokaryotes is limited. As a result, the production of genetically modified Tfs will require the involvement of more complex producers.

The low immunogenicity and high biocompatibility of Tf NPs allow the effective use of them as nano-cargo for targeted drug delivery. Thus, paclitaxel-loaded Tf NPs were delivered to MCF-7 cells in vitro, with the cell viability reaching 26.5% at 9000 ng/mL, which meant high cytotoxicity compared to non-targeted nanoparticles (paclitaxel-loaded human serum albumin NPs). Treatment of murine hepatic H22-transplanted tumor-bearing mice with paclitaxel-loaded Tf NPs led to the greatest tumor growth inhibition among all treated groups; specifically, the final tumor volume was only 4528 mm^3^ compared to the saline group (12,691 mm^3^) and paclitaxel group (8152 mm^3^) [67]. Like ferritin, Tf NPs have been successfully exploited as agent carriers for PTT/PDT. Wang et al. used IR-780-loaded Tf NPs for the treatment of colon cancer by phototherapy [69]. To measure relative cell viability in vitro, they compared the percentages of survived and untreated cells. In vitro phototherapy of IR-780-loaded Tf NPs in colon cancer cells (CT26) resulted in relative cell viability of less than 0.2 at 7.5 µg/mL of Tf NPs under laser exposure in contrast to ≥0.6 of the control group. The tumor growth was significantly inhibited in CT26-bearing mice treated with these nanoparticles under light irradiation, because for 16 days tumor volume remained almost the same as before treatment [69]. Another research group demonstrated IC_50_ of hypericin-loaded Tf NPs equal to 5 µM for treated HCT116 cells under light irradiation [70].

In recent years, such nanoparticles were used as nanocarriers capable of penetration through the BBB, which is possible because of the abundance of TfR on the brain capillary endothelial cells and tumor cells [71]. A high anticancer effect was achieved by combining Tf NP-mediated delivery and PTT (Figure 3). In vitro cell viability of U87 cells reached less than 20% after incubation with indocyanine green-loaded Tf NPs combined with laser irradiation (irradiation time is 10 min). In vivo treatment resulted in significant tumor growth inhibition with tumor volume less than 250 mm^3^ compared to nearly 2000 mm^3^ of the control group [72].

### 2.3. Targeted Encapsulin Nanoparticles

Encapsulin in a self-assembling protein nanocompartment originating from prokaryotes. Encapsulin from *Thermotoga maritima* (31 kDa monomer) has been used as a targeted drug vehicle. In 2008, *T. maritima* encapsulin, for the first time, was shown to form 24 nm spherical nanoparticles consisting of 60 monomers [73].

Successful use of encapsulin for in vitro targeted drug delivery and imaging has been recently demonstrated [74,75,76,77]. Encapsulin has a flexible surface loop and this loop can be used for the insertion of small peptides using genetic engineering techniques. The loop can be genetically modified to display a peptide tag (such as SpyTag) for a post-translational protein ligation reaction with its protein partner (SpyCatcher). The SpyTag/SpyCatcher ligation system is further discussed in detail in Section 2.12.2. With the desired modification of SpyCatcher and Encapsulin-SpyTag, such “molecular superglue” would find versatile applications: two-step targeted drug delivery, targeted bioimaging, enzyme immobilization, etc. [75]. The encapsulin’s loop can be genetically modified, e.g., to display SP94, a target peptide binding to hepatocellular carcinoma cells (Figure 4). These nanoparticles were conjugated with a prodrug, aldoxorubicin (SP94-Encap-AlDox). To confirm targeting ability, SP94-modified encapsulin was conjugated with fluorescein-5-maleimide possessing fluorescent properties. Confocal microscopy showed that these nanoparticles were specifically bound to HepG2 cells. In vitro cytotoxicity of SP94-Encap-AlDox was found to be, in a dose-dependent manner, as free Dox [74]. In another work, the encapsulin surface was functionalized with the anti-HER2 designed ankyrin repeat protein (DARPin9.29). Obtained nanoparticles were loaded with a mini Singlet Oxygen Generator (miniSOG) that produces the cytotoxic reactive oxygen species under blue light irradiation [78]. After incubation of the nanoplatforms with SK-BR-3 cells in the presence of light, 48% of apoptotic cells were identified by flow cytometry [76]. Considering all applications mentioned above, encapsulin as a targeted drug delivery platform anticipates highly promising efficacy in vivo.

### 2.4. Targeted Casein Nanoparticles

Casein is the collective name for milk proteins. It has four components: αs1-, αs2-, β-, and κ-casein, which vary in amino acids, location of hydrophobic and hydrophilic regions, and phosphorus and carbohydrate content. Normally, casein molecules are insoluble in water and, due to their amphiphilic properties, in suitable conditions, self-assemble into spherical micelles from 50 nm to 5 µm in diameter. To obtain soluble acid casein, it is converted into salt–caseinate.

The first works on the encapsulation of drugs in casein microparticles were performed in the late 1980s [79,80]. Casein has hydrophobic regions that make it a prospective agent for encapsulating other hydrophobic substances. The first report using casein micelles as a nanocapsule for lipophilic nutraceuticals was published in 2007 [81]. Since then, a series of studies concerning the encapsulation of hydrophobic compounds in reassembled casein micelles using different approaches have been published, including the following concepts of targeted casein-based drug delivery systems. Since no casein nanoparticles with genetically encoded recognizing modules have been created to date, here we highlight casein nanoparticles with a chemically modified surface for targeted drug delivery.

Many anti-cancer drugs are hydrophobic; hence, casein micelles are a prospective platform for the enhancement of drug accumulation in tumors and lowering of side effects. For example, alginate-coated caseinate NPs doubled doxorubicin delivery to the tumor site compared to freely administered doxorubicin and improved survival of Ehrlich ascites carcinoma-bearing mice by 20% compared to the control group [82]. In another study, menthol-modified casein nanoparticles loaded with 10-hydroxycamptothecin were used for glioma targeting therapy. Confocal microscopy showed a significant increase in the penetration of menthol-modified casein nanoparticles into tumor spheroids of the C6 cell line compared with bovine serum albumin nanoparticles. The in vivo efficacy of such nanoparticles was confirmed with C6 glioma treatment, and the survival time of treated mice was two times better than that of the control group [83].

Glutamic acid functionalized casein–calcium ferrite magnetic NPs co-loaded with thymoquinone were used to target U87 cancer cells. Cytotoxicity assay exhibited a 60-fold increase in the efficacy against U87 cells of nanoformulation in comparison to free thymoquinone [84].

Along with amphiphilicity and availability, casein is a biocompatible protein and is widely used for oral drug delivery [85,86,87,88]. Although casein is a prospective agent for targeted drug delivery, it is more often used for non-targeted delivery of drugs, especially anti-cancer hydrophobic chemotherapeutics and smaller degradable nanoparticles such as iron oxide NPs. Casein micelles prolong their lifetime and improve the bioavailability of encapsulated substances, thus making this protein a very promising platform for the creation of new generation drug formulations [89,90,91,92].

### 2.5. Targeted Albumin Nanoparticles

Albumin-based nanoparticles have been demonstrated to be an effective drug delivery system [93]. Albumin from different sources is used commercially for an incredibly large number of applications. Human serum albumin (HSA) is a protein with a molecular weight of 66.5 kDa. It is produced in the human liver at 10–15 g per day and its average half-life in blood serum is 19 days. Bovine serum albumin (BSA) is a protein with a molecular weight of 69 kDa, which is widely commercially available and demonstrates a high capacity to bind various substances. Although BSA is easier and cheaper in production than HSA, it can induce unfavorable immunogenic reactions after IV injection. Albumin demonstrates unique stability under a wide range of conditions including pH, high temperatures, and organic solvents. It can form nanoparticles from 20 to 1000 nm and encapsulate hydrophobic drugs prolonging its half-life time which makes it a prospective carrier of anti-cancer drugs. For instance, Abraxane, the paclitaxel-HSA nanoformulation with an average particle size of 130 nm, which was prepared through high-pressure homogenization technology, has already been approved by the U.S. Food and Drug Administration (FDA) for the treatment of several types of cancer [94,95].

Albumin itself can bind to receptors overexpressed by cancer cells of particular types, such as the 60 kDa glycoprotein (gp60) receptor and secreted protein acidic and rich in cysteine (SPARC) [96,97]. For example, nanoparticle albumin-bound (nab)-paclitaxel is shown to prolong retention of the drug in SPARC-expressing sarcoma xenografts. Nab-paclitaxel accumulation in the tumor after 6 h was 3-fold higher compared to free paclitaxel and the time of survival of mice was doubled [98].

From the late 1980s, when HSA was, for the first time, obtained from bacteria [99], many works on genetic fusions of albumin with drugs and/or targeting molecules were carried out, but no self-assembling nanoparticles were reported [100,101,102,103]. In 2020, a research group focused on the expression of fusion proteins in methylotrophic yeast *Pichia pastoris* and obtained self-assembling NPs from genetically modified HSA. Albumin was modified by introducing polyhistidine (pHis), matrix metalloproteinase-2 (MMP-2) digestion, and Arg-Gly-Asp (RGD) peptide at the separated end of HSA, using standard genetic engineering manipulations. The resulting NPs were successfully loaded with the hydrophobic drug paclitaxel and actively targeted the tumor through α_ν_β_3_-integrin up-regulated on tumor vasculature endothelium. At the tumor site, MMP-2 cutting occurs and a positively charged histidine micelle with paclitaxel penetrates deeper inside the tumor tissue. Finally, the drug is released in response to pH, as illustrated in Figure 5. The tumor growth inhibition curve for mice treated with such albumin-based NPs was similar to that of Abraxane, showing significant tumor growth inhibition efficacy with, respectively, 97.5% and 93.7% reduction of the tumor volume on day 21 [104].

Along with genetic modifications, a wide range of targeted albumin nanoparticles loaded with anti-cancer drugs are designed using the chemical modifications of amino- and carboxylic groups. For instance, the self-assembly of chemically modified albumins was shown. Namely, HSA modified with RGD peptide or photosensitizer chlorin e6 was shown to form nanoparticles capable of encapsulating paclitaxel [105]. HSA nanoparticles were conjugated with anti-α_v_ integrin antibodies and these modified nanoparticles were loaded with doxorubicin for selective targeting of αvβ3 integrin-positive melanoma cells [106]. Folate was used to obtain targeted gemcitabine-BSA [107], bexarotene-BSA [108], and artemether-HSA nanoparticles [109] for breast cancer treatment. Brain-targeted delivery of temozolomide was performed by hyaluronic acid conjugated albumin nanoparticles. The accumulation of temozolomide loaded into NPs in the brain was 8-fold greater than that of the free drug [110]. Summarizing the above, albumin is one of the prospective proteins for biomedical applications because of its availability, biocompatibility, long circulation time, capability of the encapsulation of lipophilic drugs, and potential for genetic or chemical modifications [96,111,112,113,114,115,116,117,118].

### 2.6. Self-Assembling Immunoglobulin Nanoparticles

Immunoglobulins (antibodies) are proteins used by the human immune system to neutralize different toxins and other foreign objects. Immunoglobulin G consists of four polypeptide chains and has a molecular weight of about 150 kDa. There are many commercially available IgG antibodies to various antigens. One of the best-known antibodies is trastuzumab, which recognizes the receptor HER2 which is overexpressed in some breast cancer cells [119,120]. Due to their ability to target malignant cells or pathogenic agents, immunoglobulins could be a well-suitable platform for the design of targeted nanoparticles in case the protein’s targeting abilities remain intact during NP formation.

The swift thermal formation method enables the synthesizing of NPs from immunoglobulins retaining their specificity and affinity. These nanoparticles are biocompatible and capable of encapsulating other substances, such as nanosized magnetite. Using this method, nanoparticles with various portions of trastuzumab (from 10 to 100%) were obtained [121]. Imaging flow cytometry assay showed that even NPs with 10% of trastuzumab specifically bind SKBR-3 cells overexpressing receptor HER2 [121].

### 2.7. Targeted Lactoferrin Nanoparticles

Lactoferrin (LF), a natural 77–80 kD iron(III)-binding cationic glycoprotein, consists of a single-chain backbone folded into two globular domains. It was first discovered in human milk and later reported to be found in other exocrine fluids as a regulator of free Fe(III) concentration [122]. Lactoferrin nanoparticles are considered to be promising nanocarriers for drug delivery owing to their net positive charge in physiological conditions (pI ~ 8.0–8.5) and pH-dependent release profile [123]. LF nanoparticles demonstrate a great potential for active targeting of tumor by binding to LF receptors (LFR) overexpressed on cancer cells [124]. High lactoferrin affinity to LFR also facilitates overcoming several physiological barriers such as the gastrointestinal barrier and blood–brain barrier via receptor-mediated transcytosis.

The formation of lactoferrin nanoparticles for doxorubicin targeted delivery to hepatocellular carcinoma cells was first reported in 2012 [125]. Oral administration of LF-Dox nanoparticles in hepatocellular carcinoma-bearing rats resulted in a significant decrease (>70%) of neoplastic nodules in the liver of the group treated with Dox-LF nanoparticles compared to free Dox-treated rats. Dox-LF nanoparticles also showed enhanced specificity and circulation time: area under curve (AUC) for a drug concentration in the tissue vs. time graph was enhanced in the case of blood (>100%) and liver (>50%) when administered with nanoformulations, but was drastically reduced by 4–10 fold in all the remaining tissues.

Along with cancer cells, LF receptors are overexpressed on the surface of the brain endothelial cells, allowing LF nanocarriers to pass across the BBB via receptor-mediated transcytosis. Kumari et al. demonstrated the advantages of an LF nanocarrier-based approach in overcoming the BBB for efficient brain delivery [126]. In that study, LF nanoparticles loaded with temozolomide (TMZ) were constructed for effective glioma treatment. LFRs are known to be overexpressed in glioma cells, thus allowing LF nanocarriers to accumulate in the brain tumor. LF-TMZ nanoformulations exhibited higher cytotoxicity compared to free TMZ, showing a 10-fold decreased IC_50_ value for GL261 cells (94.3 ± 2.3 and 9.3 ± 1.3 μg/mL, respectively). LF-TMZ also exhibited higher transcytosis rates across BBB in both healthy as well as glioma-bearing mice (in healthy mice, LF-TMZ concentration was more than 3 times higher compared to free TMZ 24 h post IV injection). Survival analysis has shown an increase in median survival in the mice treated with TMZ-LF NPs compared to TMZ-treated mice (25 and 18 days, respectively).

In another approach, LF-coated nanomaterials were used for targeted drug delivery to brain cells. In particular, LF-coated nanoparticles exhibited a high preference in treating Parkinson’s disease (PD) in comparison to other nanoformulations, such as PEG-PLGA nanoparticles [127]. Dopamine-loaded borneol and lactoferrin co-modified nanoparticles (LF-BNPs) prepared for this study have shown more than 4 times higher brain concentration at 0.25 h post intranasal administration compared to dopamine-loaded PEG-PLGA nanoparticles. A pharmacodynamics study has shown that the number of apomorphine-induced contralateral rotations per 15 min in 6-hydroxydopamine-lesioned rats was significantly decreased on day 20 in LF-BNPs-treated rats compared to dopamine-loaded PEG-PLGA-treated rats (>30% decrease). These facts reveal enhanced dopamine delivery with LF-BNPs into the brain for the treatment of PD. Authors explain the increased efficiency of LF and borneol co-modified nanoparticles by three different routes: an absorption pathway with subsequent crossing of the BBB, the direct pathway from nasal mucosa epithelium into the brain mainly along the trigeminal or olfactory nerves that bypass the blood–brain barrier. Since the mechanisms of substance transport through the BBB via the intranasal route of administration are not as effective as expected [128,129,130,131], the most possible mechanism of the enhanced efficiency of LF-BNPs is related to targeting of the nasal mucosa epithelium, since LFRs are overexpressed on the apical surface of respiratory epithelial cells and in the capillaries and neurons related to neurodegenerative diseases [127].

In another study, LF nanoparticles were designed for targeted delivery to inflammatory macrophages. LF nanocarriers were loaded with disulfiram (DSF) for treating inflammatory diseases [132]. LF itself possesses anti-inflammatory activity and, in combination with disulfiram, exhibits remarkable therapeutic effects on lipopolysaccharide (LPS)-induced sepsis and ulcerative colitis. Moreover, LF specifically binds with the low-density lipoprotein receptor-related protein, LRP-1, thus being a suitable targeting molecule for drug delivery to inflammatory macrophages overexpressing LRP-1. Notably, C57BL/6 mice pre-treated with DSF-LF nanoparticles before intraperitoneal injection of a lethal dose of LPS injection survived with a 100% survival rate, while two of ten mice pretreated with free DSF died after 88 h. Next, the therapeutic efficacy of DSF-LF nanoparticles against dextran sulfate sodium salt (DSS)-induced colitis was studied. DSF-LF showed higher efficiency in hampering weight decrease compared to free DSF (~92% and ~97% on day 9, respectively). This proof-of-concept study allows for considering of lactoferrin nanoparticles for the treatment of inflammatory diseases after a thorough optimization of these nanoformulations with increased therapeutic capabilities.

Narayana et al. showed the increased uptake of LF nanoparticle-encapsulated drugs in retinoblastoma cancer cells compared to the free molecular drug [133]. LF nanocarriers were loaded with carboplatin (CPT) and etoposide (ETP) for targeting cancer stem cells in retinoblastoma. LF-ETP has shown 2 times higher uptake in Rb Y79 CSCs compared to free ETP; LF-CPT has shown >1.5 times higher uptake compared to free CPT.

To date, no genetically modified approaches have been used for surface modification of LF nanoparticles, albeit certain chemical modifications have shown high efficiency in targeted delivery, e.g., Senapathi et al. prepared 2-mercaptoethanesulfonate (MES)-modified LF-nanoparticles loaded with zidovudine (AZT) for specific targeting of HIV-1 infected cells [134]. Notably, LF-MES-AZT nanoparticles (IC_50_ is 15.2 nM) demonstrated significantly greater HIV-1 inhibition in SupT1 cells when compared to free AZT (IC_50_ is 46.7 nM) and LF-AZT (IC_50_ is 34.2 nM). LF-MES-AZT nanoparticles exhibited reduced toxicity to SupT1 cells (35% difference in cell survival between LF-MES-AZT nanoparticles and free AZT after 16 h incubation), which indicates pronounced biocompatibility of AZT loaded into lactoferrin-based nanoparticles of LF-MES-AZT in comparison to free AZT.

### 2.8. Targeted Lumazine Synthase Nanoparticles

Lumazine synthase (LS) is a cage protein found in plants, fungi, and various microorganisms and is involved in riboflavin (vitamin B2) synthesis [135,136]. Depending on the organism from which it is extracted, LS may exist in different quaternary states, including pentamers and decamers, but LS derived from hyperthermophile bacteria *Aquifex aeolicus* is of the greatest interest because it forms an icosahedral capsid-like quaternary structure composed of 60 subunits (for the first time shown in 2001) with 15.4 nm outer and 9 nm inner diameters and can be used as a platform for targeted delivery (Figure 6) [135,137,138,139].

Ra et al. suggested using LS as an antigen delivery system. LS was genetically modified to carry OT-1 and OT-2 antigenic peptides and induced antigen-specific proliferation of CD4+ and CD8+ cells in vitro and in vivo [138].

Min et al. demonstrated the versatility of LS as a platform for targeted delivery. Specifically, two different targeted peptides (RGD4C and SP94 specific to pathologically activated endothelial cells and hepatocellular carcinoma cells, respectively) were genetically incorporated into the structure of LS, and which then were loaded with aldoxorubicin and bortezomib anticancer drugs. The resulting nanostructures selectively bind to target cells and cause their death, which was confirmed by fluorescence microscopy and MTT test [137]. Further, Kim et al. confirmed the versatility of LS by creating a universal antibody-binding nanoplatform consisting of LS fused with a polyvalent antibody. Such nanoplatforms were able to display different targeting antibodies on demand [140]. Levasseur et al. used a similar approach for the LS-13 variant (forming 360-subunits structure) fused to a polyvalent antibody for targeted delivery. The versatility of lumazine synthase as a drug carrier was also confirmed with the displaying of both TRAIL (TNF-related apoptosis-inducing ligand [141]) and anti-EGFR (epidermal growth factor receptor) affibodies on the particle surface [142]. However, despite specific recognition of target cells in vitro, immunogenicity experiments showed the formation of an antibody immune response to cage proteins in vivo [143].

Thus, lumazine synthase represents a promising platform for targeted delivery, however, immunogenicity issues need to be thoroughly addressed in further studies.

### 2.9. Targeted E2 Nanoparticles

The E2 protein (dihydrolipoyl transacetylase or dihydrolipoamide acetyltransferase or DLAT) is part of the pyruvate dehydrogenase multienzyme complex. It was first obtained from the bacterium *Geobacillus stearothermophilus* [144]. The E2 protein (41 kDa) [145] forms nanoparticles with a diameter of 26.6 ± 0.6 nm [146], which consist of 60 subunits. The spherical structure of the E2 protein was first demonstrated in 2005 [145].

This protein has relatively recently begun to be considered as a platform for therapeutic applications [147]. Namely, the possibilities of using these protein nanoparticles as a vaccine vehicle and in antibody-mediated responses against HIV antigens were investigated, and the ability of nanoparticles to induce antigen-specific cytotoxic T-lymphocyte (CTL) responses in vivo to model antigens was found [148,149].

Such nanoparticles have been used for targeted delivery to inflammatory breast cancer cells SUM149 in vitro [150]. To achieve the targeted delivery of nanoparticles, two components were designed to perform self-assembly through the SpyTag/SpyCatcher system. The GE11 peptide, capable of selectively binding the EGFR receptor, was used as the targeting molecule [151]. Thus, the cells were pre-targeted with GE11-mCherry-SpyCatcher fusion protein and then labeled with E2 nanoparticles fused with SpyTag with subsequent confocal microscopy study. The particles were loaded with doxorubicin and the cytotoxicity tests confirmed that doxorubicin loading into the nanoparticles is more efficient than the delivery of free doxorubicin [152].

### 2.10. Targeted Magnetosomes

Magnetotactic bacteria are gram-negative motile bacteria with the ability to form specific organelles—magnetosomes, consisting of magnetic nanoparticles enclosed in a membrane. Bacterial magnetosomes were first identified by Richard P. Blakemore in 1974 [153]. The average size of magnetosomes is 35–120 nm. The shape size and structure of magnetosome crystals varies depending on the type of bacteria [154]. Magnetosomes contain magnetic nanocrystals surrounded by a biological membrane [155,156]. The main components of the magnetosome membrane are lipids (mainly phospholipids) and various proteins.

The molecular mechanisms of magnetosome formation have not been fully addressed, but proteins associated with magnetite crystals have already been found. Mms5, Mms6, Mms7, and Mms13 are proteins that are specifically localized on the magnetic particles in the AMB-1 strain of *Magnetospirillum magneticum*. These proteins are of special interest because they play a major role in magnetosome formation, as well as in the regulation of their growth and formation rate. In particular, Mms6 is a small amphiphilic protein that self-assembles into protein micelles with N-terminal domains packed into a hydrophobic core, while hydrophilic C-terminal domains remain free and participate in nuclei formation of magnetite crystals [157,158].

Previously, molecular modeling and NMR spectroscopy showed that the DEEVE motif of the C-terminal region of Mms6 efficiently binds iron ions [159]. Thus, Mms6 is a key protein involved in the initiation of magnetosome formation by magnetotactic bacteria.

The main application of the Mms6 protein is the regulation of the growth rate and size of formed magnetite particles in biomedical applications [160]. For example, by changing the kinetics of magnetite crystal formation it is possible to obtain particles with different magnetic moments, which can be used for MRI imaging [161,162]. In addition, the use of Mms6 makes it possible to obtain magnetite of the required size without organic solvents and high temperatures, unlike conventional chemical synthesis [163]. Moreover, it was shown that Mms6 mediates the formation of cobalt ferrite nanoparticles (CoFe_2_O_4_), which are not found in living organisms [164].

Such unique properties of Mms6 allow for utilizing this protein for the creation of self-assembling targeted nanoparticles with a magnetite core without chemical conjugation steps. For example, Mms6 mediated the formation of HER2-targeted magnetite nanoparticles through the two-step delivery to HER2-overexpressing cancer cells through the proteinaceous barnase*barstar interface. Biocompatible magnetite nanoparticles were synthesized by the addition of a Bs-C-Mms6 conjugate to uncoated magnetite [165,166]. Bs-C-Mms6 contains the C-terminal part of Mms6 fused with barstar. Next, the authors used the fusion protein of barnase (binds to Barstar) and DARPin9.29 (recognizes the HER2 oncomarker) for targeted self-assembly of modified magnetic nanoparticles on the cancer cell surface. Such particles have been shown to bind effectively to the HER2 oncomarker and can be used to detect HER2-positive cancer cells in vitro.

Together with targeted delivery, magnetosomes are currently used without any specific modifications or using chemical conjugation techniques to target cancer cell surfaces. For example, magnetosomes were incubated with MDA-MB-231 cancer cells and it was shown that magnetic hyperthermia, namely, the irradiation of the cell suspension with an alternating magnetic field (at 183 kHz and field strengths of 20, 40 or 60 mT), destroyed up to 100% of cancer cells [167]. Moreover, the treatment of a xenotransplanted mouse subcutaneous tumor showed complete remission after intratumoral injection of 1 mg of magnetosomes and irradiation with an alternating magnetic field.

In another study [154], the authors compared the anticancer efficacy of two types of nanoparticles: iron oxide nanoparticles and bacterial magnetosomes coated with human serum albumin. Two types of nanoparticles were conjugated with fluorescently labeled antibodies against the EGFR receptor for targeting the MDA-MB-231 cancer cell line. It was shown that magnetosomes were more effective in terms of cancer cell binding compared to iron nanoparticles. In particular, the cellular binding rate was 92 ± 0.2% for 250 μg/mL of magnetosomes and 65 ± 5% for the same concentration of iron oxide nanoparticles. The authors also showed that magnetosomes are more effective for tumor imaging than in vivo iron oxide nanoparticles.

Magnetosomes were also successfully modified with anthracycline molecules, showing a pronounced anti-cancer effect for tumor treatment in vivo. The anticancer efficacy of doxorubicin-bacterial magnetosomes (Dox-BM) and daunorubicin-bacterial magnetosomes (Dau-BM) was evaluated using BALB/c nude mice bearing HepG2 tumor xenografts [168]. The average tumor size in the control group was >1000 mm^3^, while in mice treated with Dox-BM and Dau-BM the average tumor volume was 583 mm^3^ and 475 mm^3^, respectively. In another study [169] magnetosomes were modified with immune molecules, TGF-β inhibitor and PD-1 antibody. The resulting nanoparticles were used for the induction of ferroptosis and immunomodulation synergism in vivo on B16F10-xenograft tumor models in mice.

Currently, magnetosomes are mainly used for passive delivery to tumors. The magnetosome membrane protein Mms6 is a promising agent in anticancer therapy: modification of the C-terminal region of Mms6 makes it possible to insert polypeptide sequences of targeted anticancer molecules. In addition, the magnetosome assembling process could be regulated using Mms6. Thus, genetically encoded magnetosomes have many biotechnological applications for targeted cancer therapy, including localized drug delivery, tumor monitoring, and even magnetic hyperthermia.

### 2.11. Targeted Virus-like Nanoparticles

Virus-like particles (VLP) are 20–500 nm non-infectious nanostructures with a capsid-like morphology built from viral structural proteins. This type of nanoparticle structure was first identified in 1968 in the sera of patients with Down’s syndrome, leukemia, and hepatitis [170,171,172].

There are different approaches to the classification of VLPs, depending on the presence of lipid envelopes (enveloped and non-enveloped) and on the form (helical, icosahedral, spherical, and complex) and the number of capsid layers (1, 2, and more). Small size, wide possibilities of surface modification, high biocompatibility, various ways of VLP production, the possibility of incorporating various chemicals inside—all these factors open up wide possibilities for the application of the virus-like particles, including the creation of vaccines, delivery of drugs, genes, proteins, and other chemicals [171,172,173,174].

Currently, a wide range of virus-based vaccines have been developed and FDA-approved, including vaccines against the SARS-CoV-2 virus, but they are outside the focus of this review, and therefore will not be discussed here. However, there are many excellent reviews shedding light on this subject [175,176,177,178,179].

In addition to the aforementioned advantages of virus-like particles for biological applications, it was shown that VLPs are capable of escaping lysosomal degradation. Together, this makes virus-like particles promising candidates for targeted delivery [173].

Thus, Hagen et al. demonstrated successful usage of a recombinant adeno-associated virus (AAV) fused with DARPin or an affibody against EGFR equipped with prodrug converting enzymes for targeting and killing EGFR-overexpressed A431 cancer cells in vitro [180]. Moreover, Münch et al. showed effective tumor targeting in vivo using AAV VP2 protein fused with DARPin against HER2 [181]. Fang et al., using JC polyomavirus virus-like particles containing the HSV-tk suicide gene, demonstrated significant inhibition of bladder cancer growth in vivo [182]. Besides artificial VLPs, using oncolytic viruses in cancer therapy was proposed. These are naturally existing or genetically modified viruses that can specifically replicate in cancer cells and kill them without affecting healthy cells [183,184].

Together with anti-cancer activities, different VLPs were developed for anti-bacterial applications. Thus, Yacoby et al. reported that A12C phages loaded with chloramphenicol retarded *Staphylococcus aureus* growth 20 times more efficiently than free drug in vitro [185].

These and other studies demonstrate the versatility of VLPs as vectors for targeted delivery. The natural origin of VLPs and the possibility of loading them with genetic material allow them to be considered one of the most advanced and effective systems for the targeted delivery of DNA, RNA, and other molecules to certain areas of an organism. However, biosafety issues are still under investigation and should be taken into consideration during the development of novel targeting systems and vaccine candidates [186]. There were certain critical negative cases in the initiation period of the virus therapy, such as leukemia-like symptoms [187], high fever, multiorgan fever, and even death [188,189]. Moreover, several virus-like particles are not suitable for multiple injections due to the enhanced immune response after repeated injections [190]. However, during past decades there have been significant improvements in the development of virus-based medications, directed toward the extensive research of the biosafety issues, which strongly increased their safety and allowed effective application in healthcare, especially in vaccine use during the COVID-19 pandemic [191,192].

### 2.12. Protein-Assisted Self-Assembly of Hybrid Nanostructures

Along with self-assembling protein nanoparticles, various proteins and peptides can act as “molecular glue” to create hybrid self-assembled nanostructures. Here, we briefly describe the most common protein-assisted systems for obtaining hybrid self-assembling nanostructures.

#### 2.12.1. Streptavidin*Biotin

Historically, the most popular pair for the creation of hybrid superstructures is the streptavidin*biotin system. Biotin is a water-soluble vitamin H with a size of 244 Da. Streptavidin is a 56 kDa homotetramer from the actinobacterium *Streptomyces avidinii* that can bind up to four biotin molecules [193]. Streptavidin*biotin binding is one of the strongest non-covalent biological interactions with K_aff_ = 10^15^ M^−1^. The streptavidin*biotin complex exhibits extremely fast kinetics of binding which is stable over a wide range of pH and temperature [194]. These features made this system one of the most popular in molecular and cell biology. Thus, e.g., Ming-HanChen et al. developed a liposome-mediated drug delivery system, which consists of streptavidin-tagged liposome and biotin-tagged immune molecules (G-CSF, CD33, CD7) [195]. The efficacy of this system was demonstrated by targeting six lines of leukemia cells in vitro and a mouse xenograft model in vivo. Namely, flow cytometry in vitro tests showed that the fluorescence intensity of cell populations labeled with anti-G-CSF, CD33, and CD7 molecules through the streptavidin*biotin system proportionally depends on the expression of G-CSF receptors, CD33, or CD7 on the cell surface, respectively. Moreover, in vivo tests showed that this binding ability was maintained in mice: a biotinylated anti-CD33 antibody was injected into the bloodstream with subsequent injections of streptavidin-tagged liposomes. These liposomes were loaded with calcein and the fluorescence of only CD33-positive cells was observed during the analysis of peripheral blood, spleen, and bone marrow. The authors of another study reported the tumor targeting of radiolabeled DOTA-biotin in TAG-72-expressing tumor xenograft models [196]. The two-step delivery system based on the fusion protein of streptavidin and the scFv of the mAb CC49 and radiolabeled DOTA-biotin was used for targeted delivery to TAG-72 adenocarcinoma cancer cells. Using this approach, the authors showed that it is possible to reduce the kidney uptake of the radiolabeled compound by 30% through succinylation of the scFv-CC49-streptavidin construct [196].

Despite the described advantages of this system, some problems, such as the steric hindrance in interaction due to the difference in components’ sizes, as well as the presence of biotin in the blood of mammals, which can interfere with the formation of streptavidin/biotin pairs, prevents the widespread development of this system for in vivo applications.

#### 2.12.2. SpyTag*SpyCatcher

Another protein self-assembly system is the SpyTag*SpyCatcher complex. It consists of a modified *Streptococcus pyogenes* surface protein domain (SpyCatcher) that binds to a cognate 13-amino acid peptide (SpyTag). Upon recognition, an isopeptide covalent bond is formed between the side chains of lysine in SpyCatcher and aspartate in SpyTag [197]. The covalent interaction of peptides is a simple and powerful tool for bioconjugation and the creation of new protein architectures and biomaterials. This system is universal since the peptide label can be genetically linked to sites in target proteins [198].

Changkyu Lee and Sebyung Kang used the SpyTag*SpyCatcher ligation system to modify albumin nanoparticles for targeted delivery to cancer cells [199]. The authors encapsulated indocyanine green into nanoparticles and conjugated the HER2-recognizing affibody molecule for cancer cell targeting using SpyTag*SpyCatcher. The delivery efficiency of the obtained nanoparticles was shown in vivo on the NIH3T6.7-allografted mice model. Furthermore, the obtained nanoparticles were used for PTT [199].

A more advanced system, namely, the third generation SpyTag003/SpyCatcher003 pair with a k_on_ = 5.5 × 10^5^ × M^−1^·s^−1^ close to the diffusion limit, has been further developed. The reaction proceeds in a few minutes even at nanomolar concentrations of both peptides [200]. In one of the latest works, the authors used the SpyTag003/SpyCatcher003 system for adenovirus genetic engineering with a plug-and-display technology based on SpyTag003/SpyCatcher003 coupling chemistry [201]. Most applications of the SpyTag003/SpyCatcher003 system are currently focused on in vitro studies, however, SpyTag003/SpyCatcher003 is a promising platform for the self-assembly of targeted peptide nanoagents in vivo.

#### 2.12.3. Barnase*Barstar

Another two-step system for creating hybrid nanostructures is the barnase*barstar complex. Barnase (12 kDa extracellular RNase from *Bacillus amyloliquefaciens*) and its 10 kDa inhibitor barstar form a complex with high affinity (K_aff_ = 10^14^ × M^−1^) [202,203,204]. At the same time, the N- and C-terminal parts remain available for further interaction with other molecules. This system is actively used for the creation of modular targeting molecules, in particular, various nanoparticles that can be functionally attached to barstar or barnase. In addition, these proteins proved to be highly biocompatible in vivo [205], thus giving rise to the development of a whole range of studies on two-step delivery.

The barnase*barstar complex is widely used for the two-step targeted delivery of nanomaterials to cancer cells in vitro and in vivo [206,207,208]. Namely, HER2- and EGFR-targeting was realized with quantum dots [206], magnetic particles [207,208], and polymer particles [208] in vitro, as well as CAR-T cells in vivo [209]. Barnase*barstar is the all-protein pre-targeting system suitable for *E. coli* production, which is extremely stable under severe conditions, such as high temperature and high concentration of the chaotropic agents [210]. N- and C-terminals of both proteins are available for chemical conjugation and genetic manipulations, thus making this pair universal for different self-assembling nanostructures for a plethora of oncotargets.

#### 2.12.4. Antibody*Hapten

Antibodies are proteins that can bind their antigen specifically and with high affinity. Dissociation constants of IgG*antigen complexes are in the nanomolar range [211], sometimes reaching picomolar regions (e.g., anti-interleukin 6 antibody towards the interleukin 6 Kd is 5 pM [212]). The high affinity of antibody*hapten complexes, as well as antibody molecule stability, allows for obtaining supramolecular structures possessing a combination of properties based on nanoparticles of different origins that are extremely stable under severe conditions [210]. The antibody*hapten complex is widely used in molecular and cell biology, ELISA assays [213], in nanomedicine for targeted delivery applications, as well as for the creation of smart nanoagents acting as biorobots according to Boolean logic sensing molecules in the microenvironment [3].

#### 2.12.5. Lectin*Glycoprotein

Lectins are naturally occurring proteins capable of specific and reversible binding to carbohydrates and their residues and thus are promising molecules for the targeted delivery of therapeutic agents into cancer cells, as their glycosylation profile may differ from that of normal cells [214,215]. Although currently known lectin*glycoprotein complexes, such as protein pairs mediating nanoparticle self-assembly, are not very popular among researchers, these protein interactions represent a promising tool for the effective and reversible assembly of nanoparticle-based superstructures. Namely, we earlier synthesized gold and magnetic nanoparticles, which were then modified with lectins or glycoproteins by adsorbing protein to the particle surface at a pH corresponding to the isoelectric point of the protein [216]. The comprehensive study on the lectin*glycoprotein interaction within the composition of nanoparticles of different origins was carried out. The specificity of the binding of nanoparticles modified by different lectins (or glycoproteins) to a wide range of glycoproteins (or lectins, respectively) was shown, thus demonstrating the most effective lectin*glycoprotein pairs for nanoparticle self-assembly, such as bovine lactoferrin and concanavalin A, or porcine gastric mucin with wheat germ agglutinin [216].

#### 2.12.6. Antibody*Protein A/G/L

Protein A is a 42 kDa cell wall protein of *Staphylococus aureus*, consisting of five homologous domains and having the ability to bind to the Fc fragment of IgG of some species (e.g., human and rabbit) with high affinity [217]. Thereby, staphylococcus protein A (SpA) is widely used in affinity chromatography and biosensors. In addition, it has an application in various immunological assays, such as radioimmunoassay, immunoprecipitation, and ELISA, due to its ability to interact with a wide range of mammalian IgG with different specificities [218]. Along with ELISA applications, SpA*IgG interaction is widely used for the development of two-step targeted systems due to the ability of oriented modification of nanoparticles with IgG molecules making Fc fragments bound to the nanoparticle surface and Fab fragments available for target binding. Namely, Gayong Shim and Dongyoon Kim developed a nanoparticle with a controlled orientation of antibodies specific to the HER2 receptor [219]. First, a PEGylated lipid was conjugated with a peptide, which is a fragment protein A that recognizes the Fc portion of an IgG, and then liposomes were obtained based on the modified lipid. These liposomes were then modified to the anti-HER2 antibody with high efficiency. The effective accumulation of targeted liposomes with non-covalently attached antibodies was confirmed in vivo using Nu/Nu mice.

Along with protein A, protein G is widely used. This 60 kDa protein was isolated from the cell wall of *Streptococcus* sp. group G and was shown to possess increased avidity for rabbit IgG in comparison with protein A. Protein G, as well as protein A, is used for sterically oriented non-covalent immobilization of tumor-targeting antibodies on the surface of various nanoparticles. Thus, Liuen Liang and Andrew Care designed upconverting nanoparticles (UCNPs) with a silica coating and encapsulated Rose Bengal photosensitizer [220]. Silica-coated UCNPs were modified with a silica-binding peptide fused with protein G and then functionalized with anti-EpCAM antibodies, thus realizing targeted delivery. Selective killing of cancer cells by generation of reactive oxygen species and NIR-triggered phototoxicity was demonstrated in vitro in human colorectal adenocarcinoma HT-29 cells, thus proving the efficacy of the designed nanoparticle self-assembly protocol [220].

Slightly less popular is protein L, which was first derived from the cell wall of *Peptostreptococcus magnus.* In contrast to Protein A and Protein G, which bind the Fc region of IgG, Protein L was shown to bind the light chain of IgG [221]. Protein L binds a wider range of IgGs than Protein A or G, but this protein is not the best choice for the mediation of two-step targeted delivery. Because it cannot provide Fc-oriented binding or recognizing proteins, and steric hindrance occurs during the target recognition, this protein is applied more widely in affinity chromatography for Fab fragments purification [222].

## 3. Discussion

The advantages of protein self-assembling nanoplatforms for biomedical applications include long blood circulation, evasion of immune responses, the absence of the accelerated blood clearance phenomenon, high biocompatibility, biodegradability, and some others. Standard genetic engineering techniques make possible simple manipulation in order to design targeted protein self-assembling nanostructures directed to different oncomarkers, such as EGFR, HER2, EpCAM, and others. The loading capacity of several types of nanoparticles allows them to be equiped with either diagnostic or therapeutic capabilities, thus realizing bioimaging and therapeutic applications, including chemotherapy, gene delivery, and photodynamic and photothermal therapy.

The field of targeted self-assembling nanoparticles is still relatively new, and today there are only a limited number of studies demonstrating targeted diagnostics and therapy for socially significant diseases with the use of such platforms and the most interesting of them are summarized in Table 1.

We believe that biocompatible and non-immunogenic self-assembling nanoparticles, such as encapsulin-based protein structures, are one of the most prospective targeted drug delivery platforms due to their thermostability and pH- and proteolysis-resistance [223,224,225]. Moreover, different genetic modifications of the encapsulin surface enabling oriented functionalization with targeted molecules have already been discovered and demonstrated for in vitro applications, which makes us believe in the development of encapsulin-based smart delivery systems over the next two decades [225,226].

However, despite the obvious advantages of protein-based systems in terms of translation into clinical practice, there are a number of significant limitations that must be taken into account during the development of new targeted delivery systems. In particular, the large-scale industrial production of protein medications is an expensive process that requires careful selection of an effective host for the protein production, which will provide proper post-translational modifications, thorough optimization of cultivation and extraction processes, as well as the development of proper purification systems and quality control tests.

The transition of protein biosynthesis and purification protocols from the standard biochemical laboratory to large-scale biotechnological production is hampered by many difficulties, however, the development of continuous bioreactors and chromatography systems within high-throughput systems allows us to establish reproducible techniques for the production of protein- and peptide-based medications. Recent developments in the industrial production of proteins have given rise to several types of protein-based drugs in the clinic, such as monoclonal antibodies (e.g., anti-HER2 IgG, trastuzumab, marketed under the trade name Herceptin, Roche, or anti-CD20 IgG, rituximab, marketed under the trade name Rituxan, MabThera), different enzymes (e.g., bovhyaluronidase azoximer marketed as Longidaze, Petrovaks), peptides (e.g., glucagon-like-1 peptide marketed as Levemir, Novo Nordisk), and human serum albumin nanoparticles (paclitaxel-loaded albumin nanoparticles Abraxane, Sanofi-Aventis), thus proving the efficiency of protein-based medications and the possibility of their large-scale production [227,228,229,230]. Moreover, several complex organic nanoparticles, such as extracellular vesicles and protein particles are already undergoing clinical trials, thus confirming the achievability of large-scale production and purification of protein, lipid, and protein/lipid nanoparticles for clinical applications [231,232,233].

However, scaling up the biotechnological production of chemically synthesized nanoparticles often leads to poor batch-to-batch reproducibility and an increased polydispersity index. Nevertheless, this problem is minimized for self-assembling protein nanoparticles: since the amino acid sequence is synthesized in a unique way according to the DNA sequence, and the self-assembly of nanoparticles is definitely controlled by external conditions, such as temperature, pressure, and pH, it is worth believing that the biotechnological production of self-assembling protein nanoparticles will lead to much more reproducible nanocarriers compared to chemically synthesized particles, such as, for example, iron oxide nanoparticles for MRI imaging.

Despite the fact that protein nanocarriers can be synthesized in bacterial producers with high reproducibility, a number of problems arise in their purification for final administration to humans. Notably, lipopolysaccharides (LPS) from bacterial cell walls can cause significant hypersensitivity reactions in humans, up to anaphylaxis-like shock, and careful removal of LPS is a serious biotechnological problem. This problem is solved by affinity chromatography, e.g., with poly(ε-lysine) derived cellulose beads, or microfiltration and ultrafiltration based on membrane adsorbers with subsequent chromogenic tests, e.g., LAL-test based on LPS measurement via the clotting reaction of the hemolymph of the horseshoe crab, Limulus amebocyte lysate [234].

As natural carriers, self-assembled protein nanoparticles exhibit unique properties for drug delivery in vivo—the therapeutic capabilities can be incorporated into the protein structure, e.g., by the creation of self-assembling proteins fused with truncated genetically encoded fragments of bacterial toxins (such as, e.g., fragments of *Pseudomonas aeruginosa* exotoxin A [119,235,236]) or cytotoxic properties can be incorporated into the nanoparticle structure via loading with a chemotherapeutic drug, such as doxorubicin or paclitaxel. The drug loading process is an additional serious obstacle in the creation of novel medical nanoformulations. Currently, different techniques are utilized for the loading of therapeutic substances into protein or lipid nanoparticles, such as the pH gradient method, mixing, freeze–thaw cycles, hypotonic dialysis, and other methods [237]. Some of these methods, such as electroporation and sonication, are very efficient but suitable only for small-scale laboratory applications. Other methods, such as hypotonic dialysis, thermal shock, and extrusion techniques, are more suitable for large-scale production, but require expensive equipment and specific consumables, thus the development of effective drug loading systems is one of the top priorities in the development of protein delivery systems for chemotherapeutic drugs [237].

Moreover, the methods of drug loading that have a physical effect on nanoparticles, such as sonication, electroporation, or freeze–thaw cycles, can lead to nanoparticle aggregation which is not acceptable for intravenous administration. One of the most important challenges is the development protein drug delivery systems that are stable after the loading with the chemotherapeutic drug and do not aggregate after this process, and it is critical for such systems to remain stable for a long time, ideally in lyophilized form at room temperature.

We do believe that lyophilization for further storage and reconstitution in aqueous buffer systems is necessary for the creation of new protein nanomedications. For most nanocarriers loaded with low molecular weight compounds, there is an effect of so-called “burst-release”, when, usually, from 5% to 30% of the loaded substance is released from the nanocarrier within a short time period through diffusion [238,239,240]. This process can be significantly slowed down by modifying the nanocarrier surface with various stabilizing polymers, such as, for example, chitosan oligosaccharide lactate, or by using a specific matrix for loading [236,239,241]. It is critical to note that “burst-release” still cannot be fully avoided but can be reduced only by completely drying the sample for further storage.

**Table 1 pharmaceutics-15-00231-t001:** Characteristics and application of targeted protein self-assembling nanoparticles.

Protein	NP Size	Targeting Molecule	Loaded Molecule	In Vitro/In Vivo Applications, the Main Result of the Study	Refs
Ferritin	12 nm	RGB peptide	Sinoporphyrin sodium (DVDMS)	In vitro—4T1 cell culture, in vivo—4T1 tumors. The relative cell viability of 4T1 cells was less than 40% and 100% tumor elimination under light irradiation is shown in 2 weeks.	[62]
Transferrin	5–200 nm	Transferrin is the targeted molecule itself binding to TfR	IR780	In vitro—CT26 cell culture, in vivo—CT26 tumors. Laser-induced CT26 cell death and tumor growth inhibition were shown.	[69]
Casein	50–500 nm	Menthol	10-hydroxycamptothecin	In vitro—C6 cell culture, in vivo—C6 glioma tumors. Cell toxicity with IC_50_ = 0.0397 µg/mL, increased mice survival rate with C6 glioma.	[83,242]
Lactoferrin	Differs significantly	Lactoferrin binds to Lf receptors	Temozolomide	In vitro—GL261 mouse cell culture, in vivo—glioma tumors. Cell toxicity with IC_50_ = 9.3 ± 1.3 µg/mL, tumor growth inhibition, and a high portion of apoptotic cells in the tumor.	[243]
Albumin	100 nm	RGD peptide	Paclitaxel	In vivo—mice with MGC-803 tumors. Tumor growth inhibition = 97.5%.	[104]
Immunoglobulin	84–150 nm	Trastuzumab	Cy3	In vitro—SKBR3 и CHO cell lines. HER2-specific cell targeting is demonstrated.	[121]
Encapsulin	24 nm	DARPin9.29	miniSOG	In vitro—SKBR3 cell line. 48% of cells treated were eliminated through the apoptosis induction.	[73,76]
Lumazine synthase	15.4 nm	anti-EGFR affibody	TRAIL	In vitro—A431 cell line, in vivo—A431 tumors. Cell toxicity with IC_50_ = 6.62 nM; two-step targeted delivery resulted in tumor growth inhibition = 70%.	[142,139]
Magnetosomes	35–120 nm	DARPin9.29	Magnetite	In vitro—SKBR3 и CHO cell lines. HER2-specific cell targeting is demonstrated.	[165,166,244]
E2	26.6 ± 0.6 nm	GE11 peptide	Doxorubicin	In vitro—breast cancer cell line SUM149. Cytotoxicity of doxorubicin loaded into the nanoparticles is more efficient than the delivery of free doxorubicin.	[145]
Bacteriophage MS2	27 nm	SP94 peptide	Doxorubicin, cisplatin and 5-FU	In vitro—hepatocellular carcinoma cells HCC. SP94-targeted MS2 nanoparticles allow the elimination of cancer cells with IC_50_ = 10–15 nM.	[245]

## 4. Conclusions

We believe that the rapidly developing field of nanotechnology will make it possible to apply all the achievements of nanomedicine to protein nanoparticle design, while standardized protocols for the production and purification of proteins will make it possible to obtain batch-to-batch reproducible samples of targeted self-assembling protein nanoparticles, which will significantly speed up the process of approving such particles for clinical applications for personalized medicine.

## Figures and Tables

**Figure 1 pharmaceutics-15-00231-f001:**
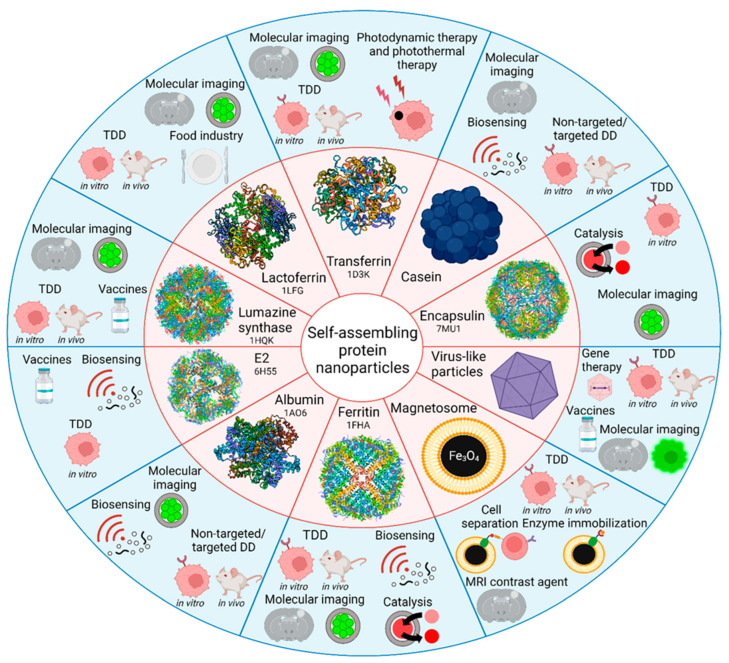
The diversity of self-assembling protein nanoparticles and their applications for in vitro and in vivo research, including biosensing, molecular imaging, and photodynamic or photothermal therapies. The structure of protein nanoparticles is accompanied by the Protein Data Bank (PDB) number if applicable.

**Figure 2 pharmaceutics-15-00231-f002:**
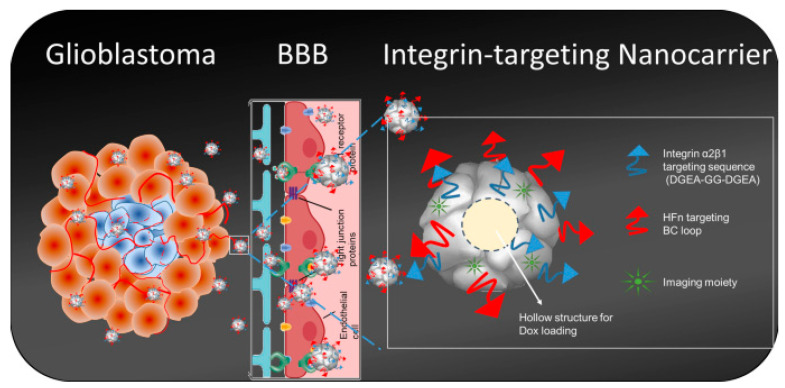
Ferritin nanoparticles for glioblastoma treatment. Targeted delivery of 2D-HFn to glioma through the BBB. Reprinted with permission [57] (CC BY 4.0).

**Figure 3 pharmaceutics-15-00231-f003:**
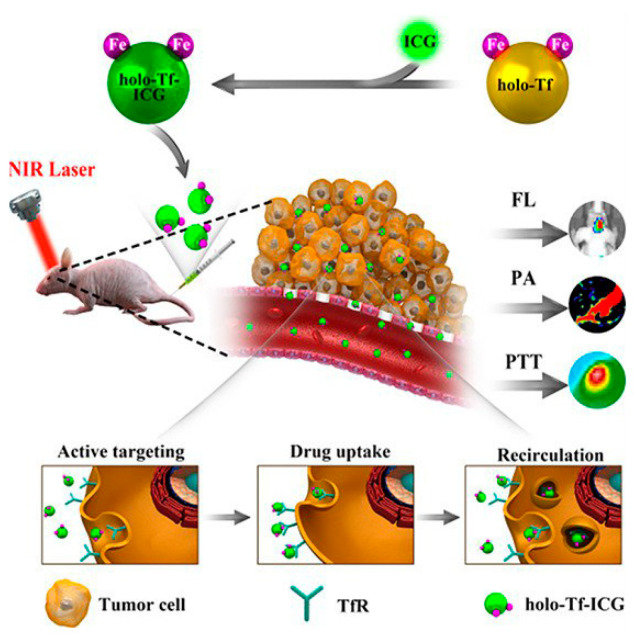
Schematic representation of Tf-NP-based targeted delivery of indocyanine green (ICG) for PTT. Reprinted with permission from [72]. Copyright 2017, American Chemical Society.

**Figure 4 pharmaceutics-15-00231-f004:**
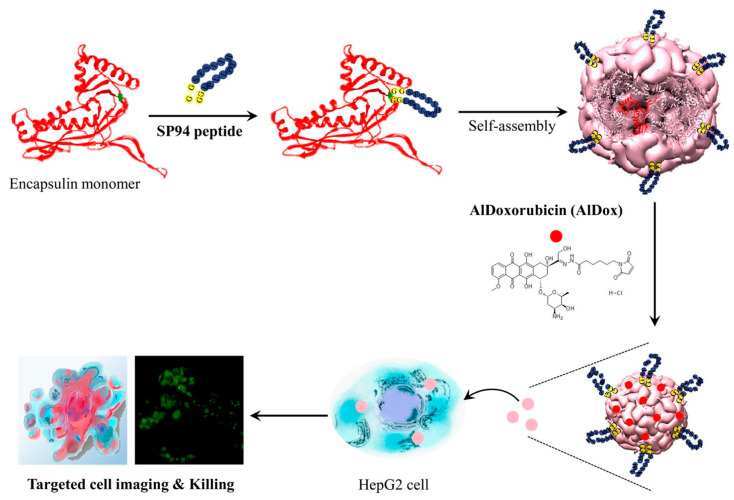
Scheme of encapsulin-based targeted drug delivery to hepatocellular carcinoma cells. Reprinted with permission from [74]. Copyright 2014, American Chemical Society.

**Figure 5 pharmaceutics-15-00231-f005:**
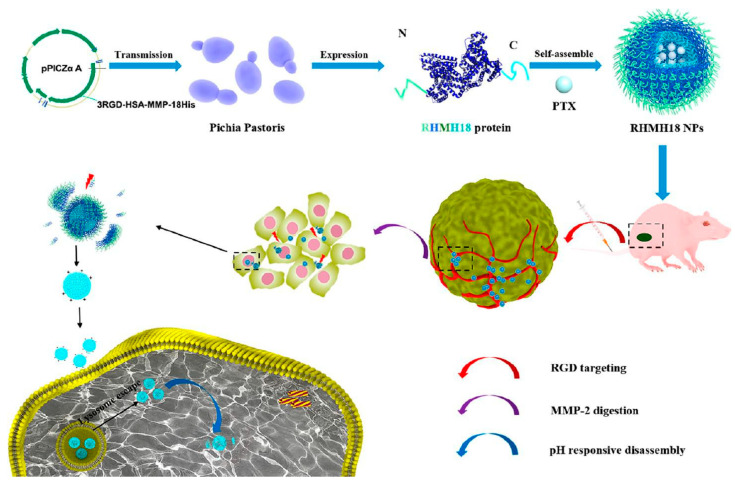
Schematic illustration of the design and application of self-assembling albumin nanoparticles for RGD-mediated tumor targeting. Copyright 2020 American Chemical Society [104].

**Figure 6 pharmaceutics-15-00231-f006:**
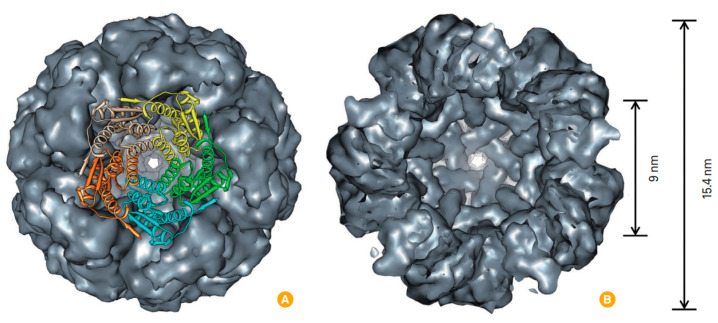
Schematic illustration of lumazine synthase nanoparticle: surface and ribbon diagram (PDB: 1HQK) looking down the five-fold symmetry axis (**A**) and the interior space of the protein cage (**B**). Reproduced with the permission from [138] (CC BY-NC 4.0).

## Data Availability

Not applicable.

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
