# Peer review of "Genetically Encoded Self-Assembling Protein Nanoparticles for the Targeted Delivery In Vitro and In Vivo"

_pharmaceutics, 2023, doi:10.3390/pharmaceutics15010231_

Round 1

Reviewer 1 Report

This is an excellent and a very much timely review. I recommend publication as is. A minor editorial comment: in several places within the text, the designation for IC50 needs to have the 50 adjusted as a subscript.

Author Response

Comment 1

This is an excellent and a very much timely review. I recommend publication as is. A minor editorial comment: in several places within the text, the designation for IC50 needs to have the 50 adjusted as a subscript.

Reply 1

We thank the reviewer for this comment, the issue on the designation for Ic50 was corrected.

Reviewer 2 Report

The review discusses a relatively new technology in the field of drug delivery providing comprehensive, updated and important data. However, the manuscript in general is too focused on the advantages and the reported activities of these nanoparticles or targeted delivery systems. There is little mentioned in the review about the limitations of these systems such as cost, side effects, difficulty of production or scaling up. Discussing these limitations is beneficial to researchers reading the review as it can guide them to possible research objectives or ideas to solve these limitations or strategies to avoid them.

The review should provide more information regarding the extraction/purification of the genetic encoded nanoparticles as well as the drug loading process as these may represent a challenge to developing delivery systems using this type of particles. Additionally, the review does not mention any details regarding the release of drugs from these nanoparticles or the stability of the formulations.

 In addition to this general comments, the following comments are recommended for consideration before the manuscript is published:

1-      The keywords: HER2, HER1 PDT and PTT seems to be irrelevant to the topic of the review. I understand that these are targets for targeted drug delivery systems, however, these are common to all delivery systems, and they are not particularly related to genetic encoded proteins considered for this review. It is recommended to remove these keywords or replace them with more relevant keywords.

2-      Minor language errors:

a.       Line 64: “studied” should be “studies”

b.       Line 121: the abbreviation “TGI” was used without prior definition in the manuscript.

c.       Line 368: “modules” should be “nodules”

d.       Line 447: redundancy in “TNF-related apoptosis-inducing that induces apoptosis”

e.       Line 509: Authors of the most recent Biocompatible magnetite nanoparticles were synthesized by the addition of Bs-C-Mms6 conjugate to uncoated magnetite [152,153], The sentence should be reviewed (Authors were synthesized)

f.        Line 535: “Boxorubicin” should be “doxorubicin”

g.       The sentence starting line 552 is repeated twice

3-      Some sentences and paragraphs in the manuscript require extensive revision to ensure clarity of the content. For example, “The flow cytometry revealed high binding affinity accounting for less than 1 nM” (Line 149).

4-      The manuscript mentions in line 65 that there is no nanoparticle formulation approved by the FDA till now. However, this does not seem to be true. The FDA has approved several nano-formulations. Please check table 1 of the following reference for examples of approved nanomedicines. There are several other references available in literature as well:

Halwani, Abdulrahman A. "Development of Pharmaceutical Nanomedicines: From the Bench to the Market." Pharmaceutics 14.1 (2022): 106.

5-      In the paragraph starting line 365 the authors mentioned that lactoferrin reduced the concentration of doxorubicin in all tissues except blood and liver. Does this include the brain as in the following paragraph the review mentions that lactoferrin enhances the delivery to the brain by acting on corresponding receptors on the BBB while free doxorubicin normally does not cross the BBB?

6-      In line 390, the review mentions that Dopamine-loaded borneol and lactoferrin co-modified nanoparticles were used for delivery to the brain for treatment of Parkinson’s. What was the advantage of these nanoparticles compared to the PEG-PLGA nanoparticles taking into consideration that the intranasal route is already known to lack the BBB structure?

7-      In 398, lactoferrin nanoparticles were used to target macrophages. Does this imply that these nanoparticles are rapidly engulfed from the circulation by macrophages? If this is true, will it limit their use for other targets? Or were these nanoparticles modified purposely in some way to target macrophages? Additionally, the differences between the treatment with the nanoparticles and the free drug treatment was minimal in both LPS inflammatory model and induced Colitis model and cannot confirm the efficiency of the nanoparticles for this route of administration

8-      The limitations related to the use of virus like particles (VLP) should be expanded to include other known limitations than biosafety. For example, the use of VLPs has been associated with leukemia like symptoms, high fever, multiorgan fever and death in some instances (Stolberg, 1999, Abina et al., 2003). Additionally, VLPs are not suitable for repeated dosing as they cause induce immune responses with repeated exposure (El-Aneed, 2004).

9-      The SpyTag/SpyCatcher system is mentioned several in the manuscript before it was discussed in details in section 2.12.2. It will be helpful for the readers to add a note when it was first mentioned that “this system will be discussed later” or is discussed in section ….. of the manuscript as it may not be familiar to all readers.

Author Response

Comment 1

The review discusses a relatively new technology in the field of drug delivery providing comprehensive, updated and important data. However, the manuscript in general is too focused on the advantages and the reported activities of these nanoparticles or targeted delivery systems. There is little mentioned in the review about the limitations of these systems such as cost, side effects, difficulty of production or scaling up. Discussing these limitations is beneficial to researchers reading the review as it can guide them to possible research objectives or ideas to solve these limitations or strategies to avoid them.

Reply 1

We thank the reviewer for this valuable comment. Indeed, the limitations of protein-based systems for targeted delivery should have been discussed. The issue is now addressed within the manuscript as follows:

“However, despite the obvious advantages of protein-based systems in terms of translation into clinical practice, there is a number of significant limitations that must be taken into account during the development of new targeted delivery systems. In particular, the large-scale industrial production of protein medications is an expensive process that requires careful selection of an effective host for the protein production which will provide proper post-translational modifications, thorough optimization of cultivation and extraction processes, as well as the development of proper purification systems and quality control tests.

The transition of protein biosynthesis and purification protocols from the standard biochemical laboratory to large-scale biotechnological production is hampered by many difficulties, however, the development of continuous bioreactors and chromatography systems within high-throughput systems allows to establish reproducible techniques for the production of protein- and peptide-based medications. Recent developments in the industrial production of proteins gave rise to several types of protein-based drugs in the clinic, such as monoclonal antibodies (e.g., anti-HER2 IgG, trastuzumab, marketed under the trade name Herceptin, Roche or anti-CD20 IgG, rituximab, marketed under the trade name Rituxan, MabThera), different enzymes (e.g., Bovhyaluronidase azoximer marketed as Longidaze, Petrovaks), peptides (e.g., glucagon-like-1 peptide marketed as Levemir, Novo Nordisk), and human serum albumin nanoparticles (paclitaxel-loaded albumin nanoparticles Abraxane, Sanofi-Aventis), thus proving the efficiency of protein-based medications and the possibility of their large-scale production [225–228]. Moreover, several complex organic nanoparticles, such as extracellular vesicles and protein particles are already undergoing clinical trials thus confirming the achievability of large-scale production and purification of protein, lipid, and protein/lipid nanoparticles for clinical applications [229–231].

However, scaling up the biotechnological production of chemically synthesized nanoparticles often leads to poor batch-to-batch reproducibility and increased polydispersity index. Nevertheless, this problem is minimized for self-assembling protein nanoparticles: since the amino acid sequence is synthesized in a unique way according to the DNA sequence, and the self-assembly of nanoparticles is definitely controlled by external conditions, such as temperature, pressure, and pH, it is worth believing that the biotechnological production of self-assembling protein nanoparticles will lead to much more reproducible nanocarriers compared to chemically synthesized particles, such as, for example, iron oxide nanoparticles for MRI imaging”.

Comment 2

The review should provide more information regarding the extraction/purification of the genetic encoded nanoparticles as well as the drug loading process as these may represent a challenge to developing delivery systems using this type of particles. 

Reply 2

The relevant information is now discussed in the manuscript as follows:

“Despite the fact that protein nanocarriers can be synthesized in bacterial producers with high reproducibility, a number of problems arise in their purification for final administration to humans. Notably, lipopolysaccharides (LPS) from bacterial cell walls can cause significant hypersensitivity reactions in humans, up to anaphylaxis-like shock, and careful removal of LPS is a serious biotechnological problem. This problem is solved by affinity chromatography, e.g. with poly(?-lysine) derived cellulose beads, or microfiltration and ultrafiltration based on membrane adsorbers with subsequent chromogenic tests, e.g., LAL-test based on LPS measurement via the clotting reaction of the hemolymph of the horseshoe crab, Limulus amebocyte lysate [232]

As natural carriers, self-assembled protein nanoparticles exhibit unique properties for drug delivery in vivo – the therapeutic capabilities can be incorporated into the protein structure, e.g., by the creation of self-assembling proteins fused with truncated genetically encoded fragments of bacterial toxins (such as, e.g., fragments of Pseudomonas aeruginosa exotoxin A [119,233,234]) or cytotoxic properties can be incorporated into nanoparticle structure via the loading with a chemotherapeutic drug, such as doxorubicin or paclitaxel. The drug loading process is an additional serious obstacle in the creation of novel medical nanoformulation. Currently, different techniques are utilized for the loading of therapeutic substances into protein or lipid nanoparticles, such as the pH gradient method, mixing, freeze-thaw cycles, hypotonic dialysis, and other methods [235]. Some of these methods like electroporation and sonication are very efficient but suitable only for small-scale laboratory applications. Other methods such as hypotonic dialysis, thermal shock, and extrusion techniques are more suitable for large-scale production but require expensive equipment and specific consumables, thus the development of effective drug loading systems is one of the top priorities in the development of protein delivery systems for chemotherapeutic drugs [235]”.

Comment 3

Additionally, the review does not mention any details regarding the release of drugs from these nanoparticles or the stability of the formulations.

Reply 3

The issue is addressed in the manuscript as follows:

“Moreover, the methods of drug loading that have a physical effect on nanoparticles such as sonication, electroporation, or freeze-thaw cycles can lead to nanoparticle aggregation which is not acceptable for intravenous administration. One of the most important challenges is the development of such protein drug delivery systems that are stable after the loading with the chemotherapeutic drug and do not aggregate after this process, and it is critical for such systems to remain stable for a long time, ideally in lyophilized form at room temperature.

 We do believe that lyophilization for further storage and reconstitution in aqueous buffer systems is necessary for the creation of new protein nanomedications. For most nanocarriers loaded with low molecular weight compounds, there is an effect of so-called “burst-release”, when usually from 5% to 30% of the loaded substance is released from the nanocarrier within a short time period through diffusion [236–238]. This process can be significantly slowed down by modifying the nanocarrier surface with various stabilizing polymers, such as, for example, chitosan oligosaccharide lactate, or by using a specific matrix for loading [234,237,239]. It is critical to note that “burst-release” still cannot be fully avoided, but can be reduced only by completely drying the sample for further storage”.

Comment 4

The keywords: HER2, HER1 PDT and PTT seems to be irrelevant to the topic of the review. I understand that these are targets for targeted drug delivery systems, however, these are common to all delivery systems, and they are not particularly related to genetic encoded proteins considered for this review. It is recommended to remove these keywords or replace them with more relevant keywords.

Reply 4

The keywords were replaced with proteins that mediate self-assembly: “transferrin; lactoferrin; casein; encapsulin; albumin; lumazine synthase”.

Comment 5

Minor language errors: Line 64: “studied” should be “studies”

Reply 5

Corrected.

Comment 6

Line 121: the abbreviation “TGI” was used without prior definition in the manuscript.

Reply 6

Corrected, the abbreviation of tumor growth inhibition is defined in the manuscript.

Comment 7

Line 368: “modules” should be “nodules”

Reply 7

Corrected.

Comment 8

Line 447: redundancy in “TNF-related apoptosis-inducing that induces apoptosis”

Reply 8

Corrected to “TNF-related apoptosis-inducing ligand”.

Comment 9

Line 509: Authors of the most recent Biocompatible magnetite nanoparticles were synthesized by the addition of Bs-C-Mms6 conjugate to uncoated magnetite [152,153], The sentence should be reviewed (Authors were synthesized).

Reply 9

Corrected.

Comment 10

Line 535: “Boxorubicin” should be “doxorubicin”

Reply 10

Corrected.

Comment 11

The sentence starting line 552 is repeated twice.

Reply 11

Corrected.

Comment 12

Some sentences and paragraphs in the manuscript require extensive revision to ensure clarity of the content. For example, “The flow cytometry revealed high binding affinity accounting for less than 1 nM” (Line 149).

Reply 12

Corrected to “The flow cytometry test confirmed the binding efficiency of this protein to cells with an affinity constant of less than 1 nM”.

Comment 13

The manuscript mentions in line 65 that there is no nanoparticle formulation approved by the FDA till now. However, this does not seem to be true. The FDA has approved several nano-formulations. Please check table 1 of the following reference for examples of approved nanomedicines. There are several other references available in literature as well:

Halwani, Abdulrahman A. "Development of Pharmaceutical Nanomedicines: From the Bench to the Market." Pharmaceutics 14.1 (2022): 106.

Reply 13

The reviewer is absolutely right that there are FDA-approved nanoparticles for clinical use, and the relevant reference was added to the manuscript. However, we did not state that there are no FDA-approved nanocarriers, we do state that there are no FDA-approved targeted nanocarriers (e.g., directed to HER1 or HER2, or EpCAM).

Comment 14

In the paragraph starting line 365 the authors mentioned that lactoferrin reduced the concentration of doxorubicin in all tissues except blood and liver. Does this include the brain as in the following paragraph the review mentions that lactoferrin enhances the delivery to the brain by acting on corresponding receptors on the BBB while free doxorubicin normally does not cross the BBB?

Reply 14

Indeed, lactoferrin is traditionally considered as an effective tool for the delivery of active substances to the brain due to receptor-mediated transcytosis. Unfortunately, the authors of the study [Golla, Kishore, et al. "Efficacy, safety and anticancer activity of protein nanoparticle-based delivery of doxorubicin through intravenous administration in rats." PloS one 7.12, 2012] evaluate only the changes in accumulation in the liver (the organ of interest with hepatocellular carcinoma), blood, spleen, kidney, and heart and did not study brain accumulation. Nevertheless, these results are quite interesting, since lactoferrin is usually considered as a tool for the delivery to the brain, but, as these authors showed, delivery to the liver can also be increased if necessary.

Comment 15

In line 390, the review mentions that Dopamine-loaded borneol and lactoferrin co-modified nanoparticles were used for delivery to the brain for treatment of Parkinson’s. What was the advantage of these nanoparticles compared to the PEG-PLGA nanoparticles taking into consideration that the intranasal route is already known to lack the BBB structure?

Reply 15

The advantage of using lactoferrin-modified nanoparticles is in the targeted delivery to nasal mucosa epithelium, and this issue is discussed in the main text as follows:

“Authors explain the increased efficiency of LF and borneol co-modified nanoparticles by three different routes: an absorption pathway with subsequent crossing BBB, the direct pathway from nasal mucosa epithelium into the brain mainly along the trigeminal or olfactory nerves that bypass the blood-brain barrier. Since the mechanisms of substances transporting through the BBB via the intranasal route of administration are not as effective as expected [128–131], the most possible mechanism of the enhanced efficiency of LF-BNPs is related to targeting of nasal mucosa epithelium since LFRs are overexpressed on the apical surface of respiratory epithelial cells and in the capillaries and neurons related to neurodegenerative diseases [127]”.

Comment 16

In 398, lactoferrin nanoparticles were used to target macrophages. Does this imply that these nanoparticles are rapidly engulfed from the circulation by macrophages? If this is true, will it limit their use for other targets? Or were these nanoparticles modified purposely in some way to target macrophages? Additionally, the differences between the treatment with the nanoparticles and the free drug treatment was minimal in both LPS inflammatory model and induced Colitis model and cannot confirm the efficiency of the nanoparticles for this route of administration.

Reply 16

This study describes the use of lactoferrin nanoparticles that target lipoprotein receptor-related protein, LRP-1, which is overexpressed in inflammatory macrophages after, e.g., LPS administration. Indeed, the results are not as impressive as expected, however, the 100% survival rate was achieved using targeted lactoferrin nanoparticles, and the use of disulfiram in molecular form did not lead to such results. This issue is discussed in the main text as follows:

“In another study, LF nanoparticles were designed for targeted delivery to inflammatory macrophages. LF nanocarriers were loaded with disulfiram (DSF) for treating inflammatory diseases [132]. LF itself possesses anti-inflammatory activity and, in combination with disulfiram exhibit remarkable therapeutic effects on lipopolysaccharide (LPS)-induced sepsis and ulcerative colitis. Moreover, LF specifically binds with the low-density lipoprotein receptor-related protein, LRP-1, thus being a suitable targeting molecule for the drug delivery to inflammatory macrophages overexpressing LRP-1. Notably, C57BL/6 mice pre-treated with DSF-LF nanoparticles before intraperitoneal injection of the lethal dose of LPS injection survived with a 100% survival rate, while 2 of 10 mice pretreated with free DSF died after 88 h. Next, the therapeutic efficacy of DSF-LF nanoparticles against dextran sulfate sodium salt (DSS)-induced colitis was studied. DSF-LF showed higher efficiency in hampering weight decrease compared to free DSF (~92% and ~97% on day 9, respectively). This proof-of-concept study allows considering lactoferrin nanoparticles for the treatment of inflammatory diseases after a thorough optimization of these nanoformulations with increased therapeutic capabilities”.

Comment 17

The limitations related to the use of virus like particles (VLP) should be expanded to include other known limitations than biosafety. For example, the use of VLPs has been associated with leukemia like symptoms, high fever, multiorgan fever and death in some instances (Stolberg, 1999, Abina et al., 2003). Additionally, VLPs are not suitable for repeated dosing as they cause induce immune responses with repeated exposure (El-Aneed, 2004).

Reply 17

These limitations are now discussed as follows:

“There were certain critical negative cases in the initiation period of the virus therapy, such as leukemia-like symptoms [187], high fever, multiorgan fever, and even death [188,189]. Moreover, several virus-like particles are not suitable for multiple injections due to the enhanced immune response after repeated injections [190]. However, during past decades there were significant improvements in the development of virus-based medications directed toward the extensive research of the biosafety issues, which strongly increased their safety and allowed effective application in health care, especially in vaccine use during the COVID-19 pandemic [191,192]”.

Comment 18

The SpyTag/SpyCatcher system is mentioned several in the manuscript before it was discussed in details in section 2.12.2. It will be helpful for the readers to add a note when it was first mentioned that “this system will be discussed later” or is discussed in section ….. of the manuscript as it may not be familiar to all readers.

Reply 18

Corrected to “SpyTag/SpyCatcher ligation system is further discussed in detail in section 2.12.2”.

Reviewer 3 Report

Better quality (Higher DPI) Images are required as the one right now is not that good. 

Author Response

Comment 1

Better quality (Higher DPI) Images are required as the one right now is not that good. 

Reply 1

We thank the reviewer for this comment, indeed the quality of the figures was not high enough, it was probably the mistake of compression. The figures were updated.

Reviewer 4 Report

The submitted manuscript entitled “Genetically Encoded Protein Self-Assembling Nanoparticles 2 for the Targeted Delivery In Vitro and In Vivo” is generally well written and has good quality.

Below is a list of minor corrections that need to be done:

Abstract, line 10-11: sentence “there are no targeted drug formulations within the composition of  nanoparticles approved by the FDA for use in the clinic” vs Introduction, line 35- 36: “However, only a limited number of drug formulations within the composition of nanoparticles have reached clinical application”  please explain?

Line 38 – reference for confirmation?

Line 62- reference for Paul Erlich?

Line 94- “present” instead of “presented”

Line 121: There is no explanation for the abbreviation TGI nowhere in the text (Tumor growth inhibition)

Line 131: There is no explanation for the abbreviation FITC (Fluorescein isothiocyanate)

Line 132: U-87MG, “cells” should be added

Line 253-254: “no studies show NPs made of genetically modified molecules of casein” The authors should additionally check the literature

Line 326-328: Reformulate the sentence ” Monoclonal antibody directed against αv integrins that was covalently coupled to HSA NPs; these particles loaded with doxorubicin enabled specific targeting of αvβ3 integrin-positive melanoma cells [97].”

Line 401: LPS abbreviation needs to be introduced - lipopolysaccharide?

Line 465: CTL responses- there is no expenation for this abbreviation - Cytotoxic T lymphocyte?

Line 470: Add explanation for abbreviation first time mentioned - EGFR receptor - epidermal growth factor receptor

Line 509: lowercase letter for Biocompatible

Line 509-511: Reformulate this sentence “Authors of the most recent Biocompatible magnetite  nanoparticles were synthesized by the addition of Bs-C-Mms6 conjugate to uncoated magnetite [152,153]”

Line 519: “the irradiation the cell suspension” is missing “of”: “the irradiation of the cell suspension”

Line 535: doxorubicin instead of вoxorubicin

Line 553-555: This sentence “Virus-like particles (VLP) are 20-500 nm non-infectious nanostructures with capsid-552 like morphology built from viral structural proteins.” is repeated twice.

Line 608: What is the result of the mentioned study? to be supplemented with the result of the mentione study, The same for the study with the reference [177].

Line 685-686: Reformulate the sentence: “The comprehensive study on the lectin*glycoprotein interaction within the 685 composition of nanoparticles of different origins.”

Line 692- 719 In the Chapter 2.12.6. Antibody*Protein A/G/L, protein L - which is in the title of the chapter, is not mentioned afterwards

Line 723: “ABC phenomenon”, include an explanation for this abbreviation - The accelerated blood clearance?

Author Response

Comment 1

Abstract, line 10-11: sentence “there are no targeted drug formulations within the composition of  nanoparticles approved by the FDA for use in the clinic” vs Introduction, line 35- 36: “However, only a limited number of drug formulations within the composition of nanoparticles have reached clinical application”  please explain?

Reply 1

The difference between these statements is in the fact that indeed, there are FDA-approved nanoparticles for clinical use, but there are no targeted nanoparticles in clinics nowadays. For example, there are no nanoparticles for HER2 targeting loaded with doxorubicin, but there are particles loaded with doxorubicin with no targeting modalities (Caelyx and Myocet).

Comment 2

Line 38 – reference for confirmation?

Reply 2

The references were added as follows:

“At the same time, more than three hundred protein-based medications have been approved for clinical use [11] (there are more than one hundred of therapeutic antibodies only [12–14]; a great variety of diagnostic antibodies, including those for COVID-19 diagnostics [15]; therapeutic proteins such as different enzymes [16]; and therapeutic peptides [11]), which indicates that proteins are easier to translate into clinical practice in comparison with nanoparticles”.

Comment 3

Line 62- reference for Paul Erlich?

Reply 3

Corrected, references were added.

Comment 4

Line 94- “present” instead of “presented”

Reply 4

Corrected.

Comment 5

Line 121: There is no explanation for the abbreviation TGI nowhere in the text (Tumor growth inhibition)

Reply 5

Corrected, the abbreviation of tumor growth inhibition in defined in the manuscript.

Comment 6

Line 131: There is no explanation for the abbreviation FITC (Fluorescein isothiocyanate)

Reply 6

Corrected, the abbreviation of fluorescein isothiocyanate (FITC) is defined in the manuscript.

Comment 7

Line 132: U-87MG, “cells” should be added.

Reply 7

Corrected.

Comment 8

Line 253-254: “no studies show NPs made of genetically modified molecules of casein” The authors should additionally check the literature.

Reply 8

Indeed, this statement was incorrect and, of course, there are different genetic modifications of casein. The aim of this statement was to point out that there are no genetic constructs combining the casein gene with the gene of targeted recognizing molecules, for example, with anti-HER2 affibody. This sentence was corrected as follows:

“Since no casein nanoparticles with genetically encoding recognizing modules were created for today, here we highlight casein nanoparticles with the chemically modified surface for targeted drug delivery”.

Comment 9

Line 326-328: Reformulate the sentence ” Monoclonal antibody directed against αv integrins that was covalently coupled to HSA NPs; these particles loaded with doxorubicin enabled specific targeting of αvβ3 integrin-positive melanoma cells [97].”

Reply 9

The sentence was reformulated as follows: “HSA nanoparticles were conjugated with anti-αv integrins antibodies and these modified nanoparticles were loaded with doxorubicin for selective targeting of αvβ3 integrin-positive melanoma cells”.

Comment 10

Line 401: LPS abbreviation needs to be introduced - lipopolysaccharide?

Reply 10

The abbreviation for lipopolysaccharides (LPS) in now introduced in the manuscript.

Comment 11

Line 465: CTL responses- there is no expenation for this abbreviation - Cytotoxic T lymphocyte?

Reply 11

Corrected to “antigen-specific cytotoxic T-lymphocyte (CTL) responses”.

Comment 12

Line 470: Add explanation for abbreviation first time mentioned - EGFR receptor - epidermal growth factor receptor.

Reply 12

Corrected.

Comment 13

Line 509: lowercase letter for Biocompatible.

Reply 13

Corrected.

Comment 14

Line 509-511: Reformulate this sentence “Authors of the most recent Biocompatible magnetite  nanoparticles were synthesized by the addition of Bs-C-Mms6 conjugate to uncoated magnetite [152,153]”.

Reply 14

Corrected to “Biocompatible magnetite nanoparticles were synthesized by the addition of Bs-C-Mms6 conjugate to uncoated magnetite [165,166]”.

Comment 15

Line 519: “the irradiation the cell suspension” is missing “of”: “the irradiation of the cell suspension”.

Reply 15

Corrected.

Comment 16

Line 535: doxorubicin instead of вoxorubicin.

Reply 16

Corrected.

Comment 17

Line 553-555: This sentence “Virus-like particles (VLP) are 20-500 nm non-infectious nanostructures with capsid-552 like morphology built from viral structural proteins.” is repeated twice.

Reply 17

Corrected, the repeated sentence was removed.

Comment 18

Line 608: What is the result of the mentioned study? to be supplemented with the result of the mentione study, The same for the study with the reference [177].

Reply 18

A detailed description of these studies is added as follows:

“The efficacy of this system was demonstrated by targeting 6 lines of leukemia cells in vitro and a mouse xenograft model in vivo. Namely, flow cytometry in vitro tests showed that the fluorescence intensity of cell populations labeled with anti-G-CSF, CD33, and CD7 molecules through the streptavidin*biotin system proportionally depends on the expression of G-CSF receptor, CD33 or CD7 on the cell surface, respectively. Moreover, in vivo tests showed that this binding ability was maintained in mice: biotinylated anti-CD33 antibody was injected into the bloodstream with subsequent injections of streptavidin-tagged liposomes. These liposomes were loaded with calcein and the fluorescence of only CD33-positive cells was observed during the analysis of peripheral blood, spleen, and bone marrow. The authors of another study reported the tumor targeting of radiolabeled DOTA-biotin in TAG-72–expressing tumor xenograft models [196]. The two-step delivery system based on the fusion protein of streptavidin and the scFv of the mAb CC49 and radiolabeled DOTA-biotin was used for the targeted delivery to TAG-72 adenocarcinoma cancer cells. Using this approach, authors showed that it is possible to reduce the kidney uptake of the radiolabeled compound by 30% through succinylation of the scFv-CC49-streptavidin construct [196]”.

Comment 19

Line 685-686: Reformulate the sentence: “The comprehensive study on the lectin*glycoprotein interaction within the 685 composition of nanoparticles of different origins.”

Reply 19

Corrected to “The comprehensive study on the lectin*glycoprotein interaction within the composition of nanoparticles of different origins was carried out”.

Comment 20

Line 692- 719 In the Chapter 2.12.6. Antibody*Protein A/G/L, protein L - which is in the title of the chapter, is not mentioned afterwards.

Reply 20

Corrected, the description of protein L was added to the text:

“Slightly less popular is protein L, which was first derived from the cell wall of Peptostreptococcus magnus. In contrast to Protein A and Protein G which bind the Fc region of IgG, Protein L was shown to bind the light chain of IgG [219]. Protein L binds a wider range of IgGs than Protein A or G, but this protein is not the best choice for the mediation of two-step targeted delivery, because it cannot provide Fc-oriented binding or recognizing proteins and steric hindrance occurs during the target recognition, this protein is applied more widely in affinity chromatography for Fab fragments purification [220]”.

Comment 21

Line 723: “ABC phenomenon”, include an explanation for this abbreviation - The accelerated blood clearance?

Reply 21

Corrected, the explanation of ABC is included as “the absence of the accelerated blood clearance (ABC) phenomenon”.

Round 2

Reviewer 2 Report

The authors addressed all comments initially provided to the first version of the manuscript. No further comments or edits are needed.